# Research on the Effect of Particle Size on the Interface Friction between Geogrid Reinforcement and Soil

**Yunfei Zhao [1,2], Guangqing Yang [3,*], Zhi Wang [4] and Shaopeng Yuan [5]**

1   School of Urban Geology and Engineering, Hebei GEO University, Shijiazhuang 050031, China
2   Hebei Technology Innovation Center for Intelligent Development and Control of Underground Built Environment, Shijiazhuang 050031, China
3   School of Civil Engineering, Shijiazhuang Tiedao University, Shijiazhuang 050043, China
4   China Railway Fifth Survey and Design Instuitute Group Co., Ltd., Beijing 102600, China
5   Bostd Geosynthetics Qingdao Ltd., Qingdao 266111, China
*   Correspondence: yanggq@stdu.edu.cn

**Abstract:** For projects such as roads and railways, different fillers are often selected, and these also relate to the area where the project is located, so the characteristics of the filling soil should be considered in the design. However, the characteristics of the soil used in geosynthetic-reinforced soil (GRS) structure design are routinely simple soil properties and are not based on testing of soil with reinforcement. In order to study the influence of fillers with different particle sizes on the interface friction characteristics between the geogrid and soil, a self-developed large-scale pull-out testing machine was used. Under the action of a normal static load, pull-out tests were carried out with different fillers, such as sand, silt and gravel. According to the test results, the greater the stress applied in the normal direction, the greater the maximum pull-out force. As for the different fillers, shear stress from material with a larger particle size, such as gravel, was larger than that of sand and silt. Finally, to reveal the pattern of how the soil particles moved during the pull-out test, from a microscopic point of view, and the effect on particle–mesh size ratio, a series of discrete element method (DEM) analyses were conducted by $PFC^{2D}$. The results indicated that a larger particle is more likely to rotate and move during the test, and this makes the interlocking effect greater between the geogrid and the soil, which leads to a larger pull-out force in the laboratory test.

**Keywords:** geogrid; interface friction; particle size; DEM analysis

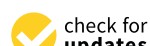



## 1. Introduction

Geosynthetic-reinforced soil walls (GRSWs) have been widely used in various engineering fields, such as roads and railways. A GRSW is a viable replacement for conventional concrete-retaining structures in infrastructure development and remedial treatments around the world. It is of great importance to investigate the mechanism of interaction between geogrid and soil in the design and application of geogrid-reinforced soil structures. There are multiple research methods to investigate the geogrid–soil interaction, and the pull-out test is the most effective one among them. A lot of research has been conducted on geogrid–soil interaction using laboratory tests.

Xiao [1], Wang [2] and Chen et al. [3] have analyzed and discussed the influence of different normal pressures, geogrid type and embedded length on the pull-out characteristics of geogrids in sand using the pull-out test. Moraci [4], Bisht [5], Alagiyawanna [6] and Teixeira [7] also studied the pull-out characteristics of the geogrid in fine-grained soil under different conditions. Wang [8] investigated the monotonic and cyclic shear behavior of the grit–geogrid interface through a series of experiments. Ding [9] and Tang et al. [10] studied the influence of the mesh size of the geogrid on the interface characteristics through the pull-out test of the geogrid in fine sand. Zhou [11] and Ezzein [12–14] used transparent fillers in a visual model box to study the microscopic mechanism of the interaction between

the geogrid and sand. However, these studies have focused on fine materials: sand. The results only indicated the pattern of interface between geogrid and sand, not suitable for other materials used in construction.

As for geogrid–soil interface characteristics with other materials such as gravel or clay, Zuo [15] and Abdi [16] performed pull-out and direct shear tests of uniaxial geogrids in sand–gravel and cohesive soils. The results showed that the shear strength of the contact surface between the geogrid and the clay was very low, but the shear strength of the contact surface with the sand–gravel material was higher, which indicated that interface characteristics with different materials would perform differently. Kim [17] conducted large-scale direct shear tests on three types of coarse-grained soil, showing that the larger the particle size, the higher the shear strength, but the tests were conducted with gravel, not other materials such as sand or silt.

Currently, research on the friction characteristics of reinforced soil has been based on laboratory tests and the results, such as cohesion and friction angle, have been macroscopic. In addition, most of these have focused on fine materials such as sand; the difference in interfacial friction properties between various particle-size materials has rarely been observed. Moreover, some studies have been focused on various mesh-size geogrids on the same particle-size materials, not on different materials. Thus, based on the laboratory tests and DEM analysis, the internal friction characteristics of reinforced soil with different particle-size materials were studied. The results may provide references for the future design and application of GRS structures.

## 2. Materials and Methods

### 2.1. Test Materials

2.1.1. Filling Materials

In order to study the interfacial friction characteristics of different particle sizes, three different filling materials were selected for testing: sand, silt and gravel. All the filling materials were collected from a local road construction site. The filling materials used in the test were subjected to geotechnical experiments to provide basic parameters for further pull-out tests and discrete element simulation studies.

Before testing, the particle gradation, specific gravity, maximum/minimum dry density and other physical indices of filling materials were measured according to the Chinese National Standard of Soil Test Method (GB/T 50123−2019).

(1)　Particle size

Based on the sieve and the densitometer methods, the physical parameters and the curve of the particle size for the three filling materials are shown in Table 1 and Figure 1.

**Table 1.** Physical parameters of the filling materials.

| Items | Sand | | | Silt | | | Gravel | | |
|---|---|---|---|---|---|---|---|---|---|
| Particle diameter (mm) | $d_{10}$ | $d_{30}$ | $d_{60}$ | $d_{10}$ | $d_{30}$ | $d_{60}$ | $d_{10}$ | $d_{30}$ | $d_{60}$ |
| | 0.16 | 0.60 | 0.96 | 0.03 | 0.13 | 0.34 | 3.2 | 3.5 | 5.0 |
| Coefficient of uniformity | 6 | | | 11.33 | | | 1.56 | | |
| Curvature coefficient | 2.34 | | | 1.66 | | | 0.77 | | |

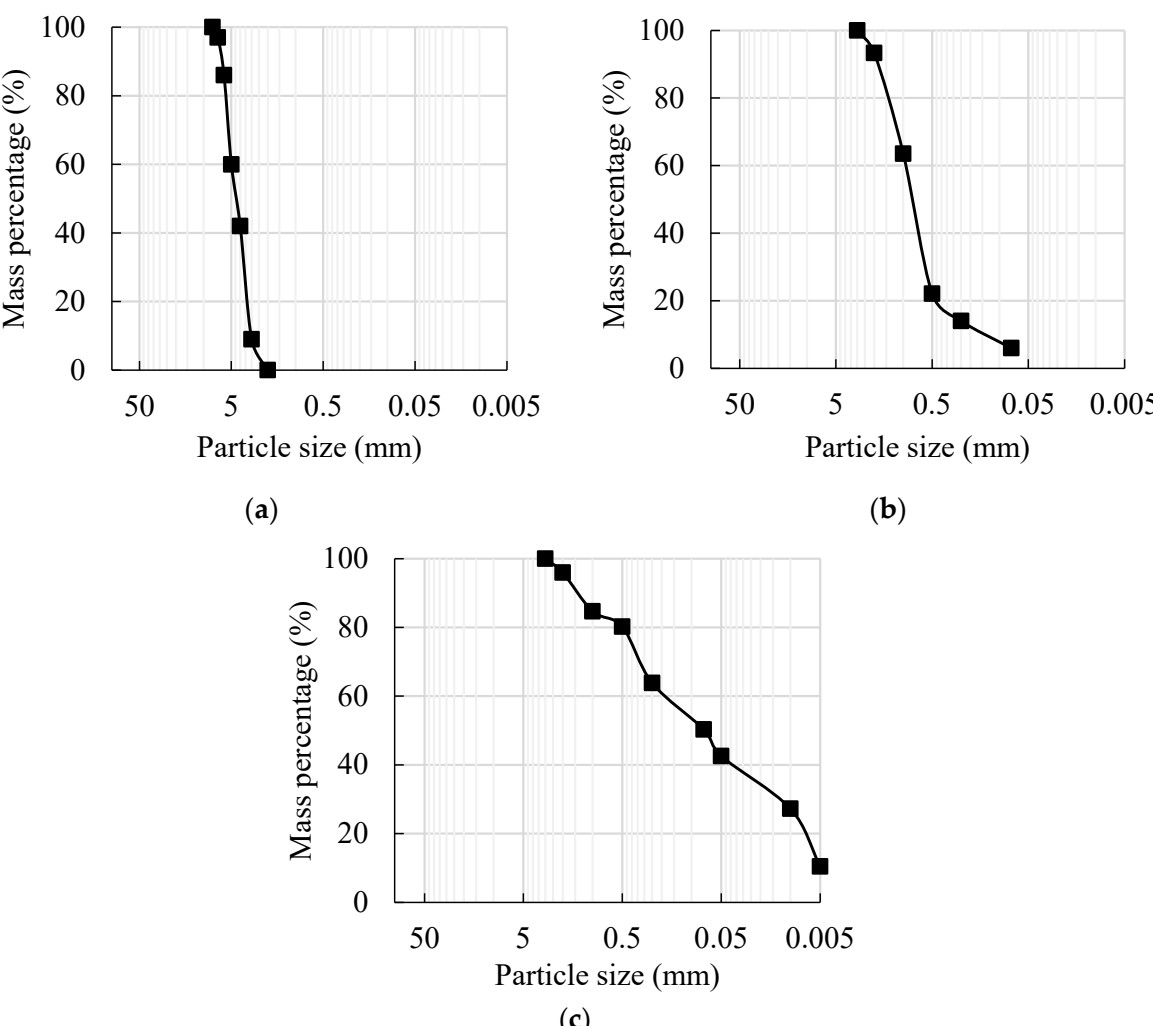

**Figure 1.** Curve of the particle-size distribution of the filling soil. (**a**) Gravel; (**b**) sand; (**c**) silt.

(2)     Compaction test

The parameters and results of the compaction test are shown in Table 2. For the sand, the maximum and minimum dry densities were 1.83 g·cm$^{-3}$ and 1.51 g·cm$^{-3}$.

**Table 2.** Parameters of the compaction test.

| Items | Indexes |
|---|---|
| Height (mm) | 305 |
| Diameter of hammer (mm) | 51 |
| Diameter of test tube (mm) | 102 |
| Hammer weight (kg) | 2.5 |
| Test tube height (mm) | 116 |
| Sample volume (cm$^3$) | 947.4 |
| Number of layers | 3 |
| Hits per layer | 25 |

The compaction test results of the three soil samples is shown in Table 3.

**Table 3.** The compaction test results.

| Sample | Maximum Dry Density (g·cm$^{-3}$) | Optimal Moisture Content (%) |
| --- | --- | --- |
| Gravel | 1.962 | 0.24 |
| Sand | 1.83 | 3.58 |
| Silt | 1.909 | 10.56 |

(3) Direct shear test

Among the important indicators that characterize soil properties are soil cohesion and internal-friction angle, which are also two extremely important parameters for subsequent establishment of discrete element models. The relationship curves between the shear displacement and shear stress of the sand, silt and gravel are shown in Figure 2.

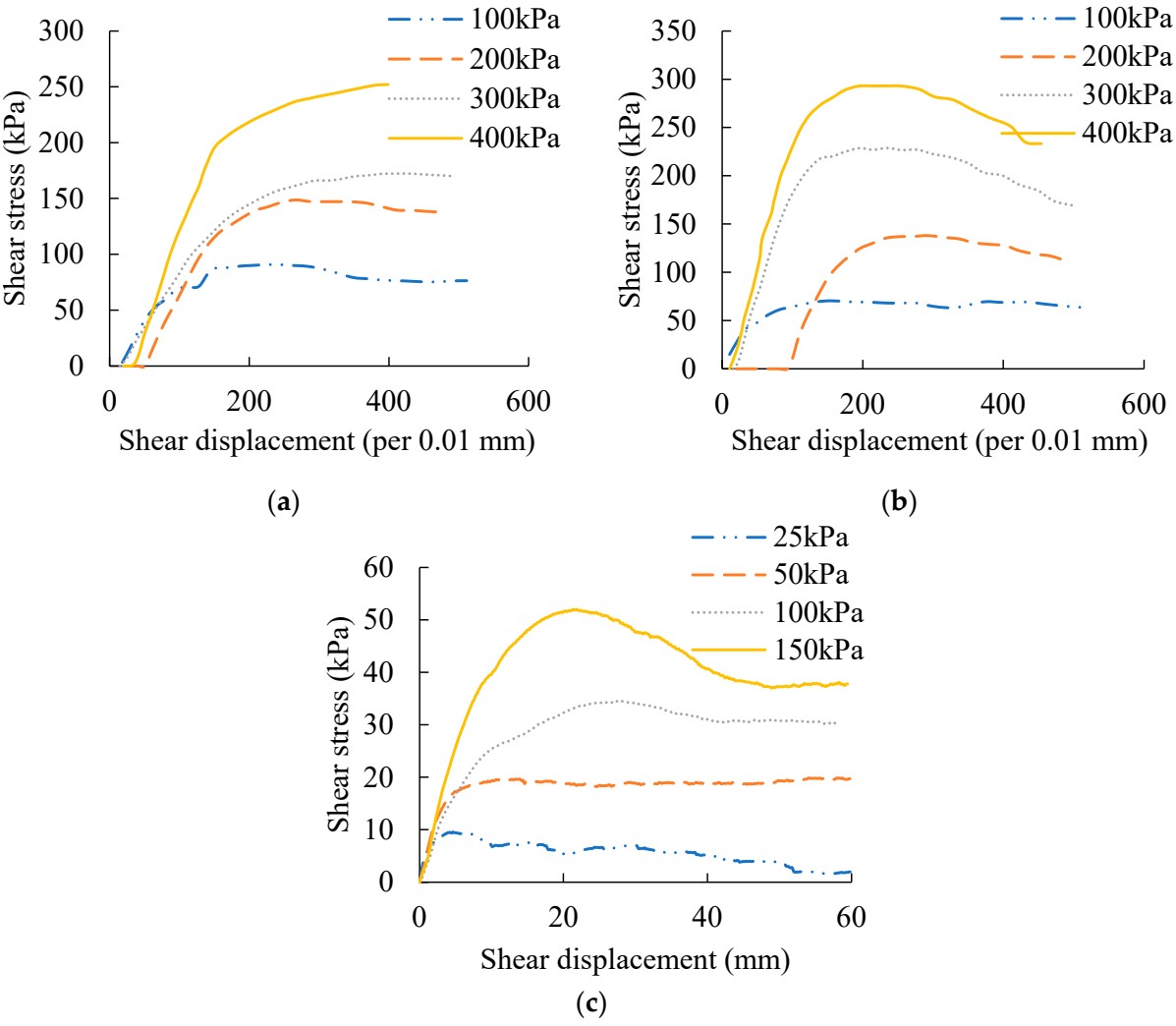

**Figure 2.** Curves of shear stress and shear displacement. (**a**) Silt; (**b**) sand; (**c**) gravel.

The direct shear test results are shown in Table 4.

**Table 4.** The direct shear test results.

| Sample | Cohesion (kPa) | Friction Angle (°) |
| --- | --- | --- |
| Gravel | 0 | 19.51 |
| Sand | 0 | 36.33 |
| Silt | 39.03 | 26.94 |

### 2.1.2. Geogrid

The geogrid was also tested by a tensile experiment, and the technical specifications are shown in Table 5.

**Table 5.** Technical specifications of the high-density polyethylene (HDPE) geogrid.

| Rib Length (mm) | | Rib Thickness (mm) | | Rib Width (mm) | | Node Thickness (mm) |
|---|---|---|---|---|---|---|
| Machine Direction | Cross Machine Direction | Machine Direction | Cross-Machine Direction | Machine Direction | Cross-Machine Direction | |
| 32 | 33 | 1.7 | 1.5 | 2.0 | 2.0 | 4.5 |

### 2.1.3. Test Equipment

The equipment used for the pull-out test of the reinforcement–soil interface was a geosynthetic material direct shear pull-out integrated tester independently developed by Shijiazhuang Tiedao University. The equipment shown in Figure 3 is mainly composed of a test box, a normal loading system, a horizontal loading system and data acquisition system. It comprises several parts of the control system. The inner diameter of the test box is $600 \times 400 \times 500$ mm (length × width × height). The vast dimensions of the pull-out box can reduce the effect of the dimensions of specimens and boundary effect for the duration of the test. In addition, to enhance the rigidity of the test box, longitudinal members made of steel were added uniformly on the outside of the pull-out box with a thickness of 10 mm to ensure the plane strain condition in the process of the test [18].

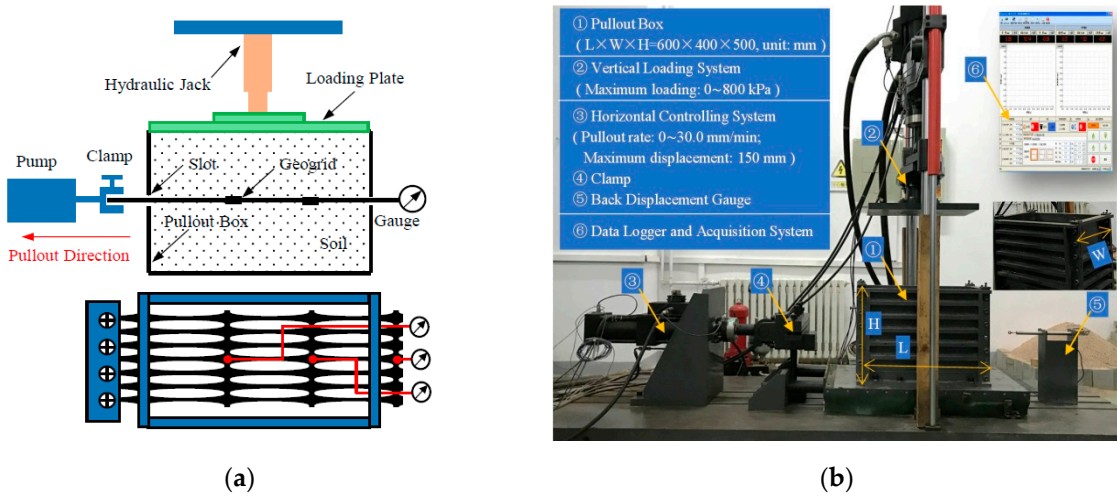

(**a**)        (**b**)

**Figure 3.** (**a**) Representation of the pull-out apparatus; (**b**) the view and technical parameters of the pull-out apparatus used in the current study.

### 2.2. Test Schemes

For all the tests, the bottom layer soil was used, having a compaction coefficient of 0.92. The density control method was used to ensure that the compactness of soil in the box met the requirements.

Three normal stresses (25 kPa, 50 kPa and 75 kPa) were chosen to simulate the tension of different embedding depths of the geogrid in actual engineering, and the drawing rate was set to 2 mm/min.

### 3. Effects of Different Fillers on the Friction Characteristics of the Interface between Reinforcement and Soil

The relationship between the pull-out force and the pull-out displacement (*s*) of the geogrid–soil interface under different normal stress conditions is shown in Figure 4. According to the test results, the pull-out force showed a linear growth trend, with an

increase in pull-out displacement under different normal stress conditions at the beginning; it then slowed down at an increasing rate and became flat or decreased after reaching the peak. This is because, after the pull-out force reached its peak, the soil particles started to rotate and move with the increasing displacement; then, the immobilization effect between the geogrid and the soil particles decreased, and most of the pull-out force became friction between geogrid and soil. Overall, the curve showed strain softening under a small load. It resembles the test results described by Zhou [11] and Ezzein [12–14].

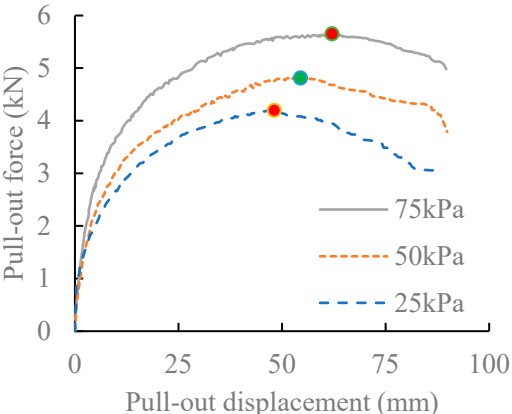

**Figure 4.** Pull-out force and displacement with sand (peaks have been marked as colored dots).

The break was set when the pull-out displacement reached 90 mm. In the pull-out tests with the corresponding normal stresses of 25 kPa, 50 kPa and 75 kPa, the maximum pull-out forces were 4.197 kN, 4.811 kN and 5.649 kN, respectively, at 48.149 mm, 53.946 mm and 62.101 mm. The maximum pull-out force increased with the increase in normal stress.

The immobilization effect between geogrid and sand was not obvious under small normal stress and the friction between those two materials was small, resulting in the maximum pull-out force being smaller in the pull-out test. Several points were randomly selected for analysis in the pull-out process, and the curve of the relationship between the pull-out force and the normal stress was plotted, as shown in Figure 5. The pull-out force increased with the increase in normal stress. The soil particles were continuously compacted and squeezed under larger normal stress; then, the immobilization effect was fully activated, resulting in the pull-out force increasing with the increase in normal stress. It can also be seen that the pull-out force increased with the increase in displacement before it reached the peak, it then went down with further displacement.

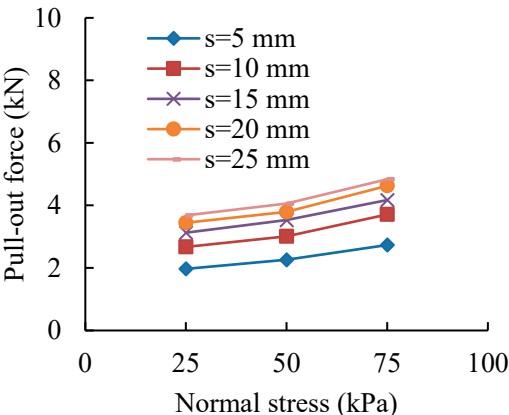

**Figure 5.** Pull-out force and stress under different displacements with sand.

The interfacial shear stress and the interfacial-friction angle were calculated, and the relationship curves were plotted according to the Mohr–Coulomb strength theory, as

shown in Figure 6. Calculated shear strength parameters are shown in Table 6. This result resembles the pattern seen elsewhere in the literature [19–22], in which the cohesion and friction angle were interface strength properties. This is where the quasi-cohesive effect between geogrid and soil was demonstrated.

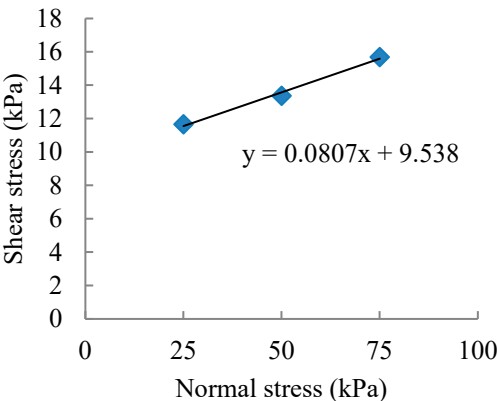

**Figure 6.** Shear and normal stress fitting curve.

**Table 6.** Pull-out test results under different normal stresses with sand.

| Normal Stress (kPa) | Maximum Pull-Out Force (kN) | Pull-Out Displacement (mm) | Shear Stress (kPa) | Interface Cohesion (kPa) | Interface-Friction Angle (°) |
|---|---|---|---|---|---|
| 25 | 4.197 | 48.149 | 11.658 | | |
| 50 | 4.811 | 53.946 | 13.364 | 9.538 | 4.614 |
| 75 | 5.649 | 62.101 | 15.692 | | |

The test results showed that the friction coefficient of the reinforcement–soil interface decreases by 44.8% with the increase in normal stress, from 0.466 at 25 kPa, 0.267 at 50 kPa to 0.209 at 75 kPa, and the reason for this was the dilatancy of the sand. Under low normal stress, the geogrid could easily have the soil particles rotating during the process of the test, during which circumstances the dilatancy of soil was obvious. With the increase in normal stress, the movement of soil particles was limited, and the soil particles could hardly rotate and move, so the dilatancy decreased. The interfacial friction coefficient was not a constant under different normal stresses (Figure 7).

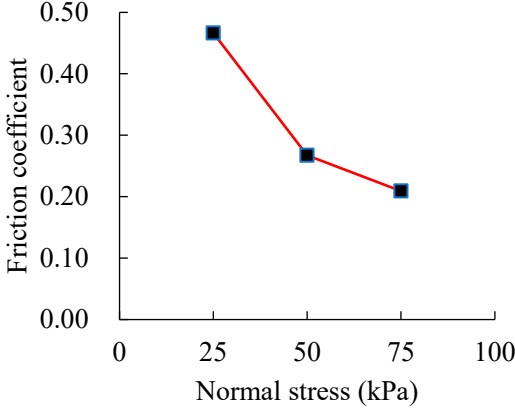

**Figure 7.** Friction coefficient under different normal stresses.

The pull-out test results of silt and gravel are shown in Figure 8. It can be seen that the pull-out curve difference of both fillers showed the same trend, as the particle size

increased the immobilization effect between the fillers and the geogrid increased with it, resulting in a gradual increase in the maximum pull-out force.

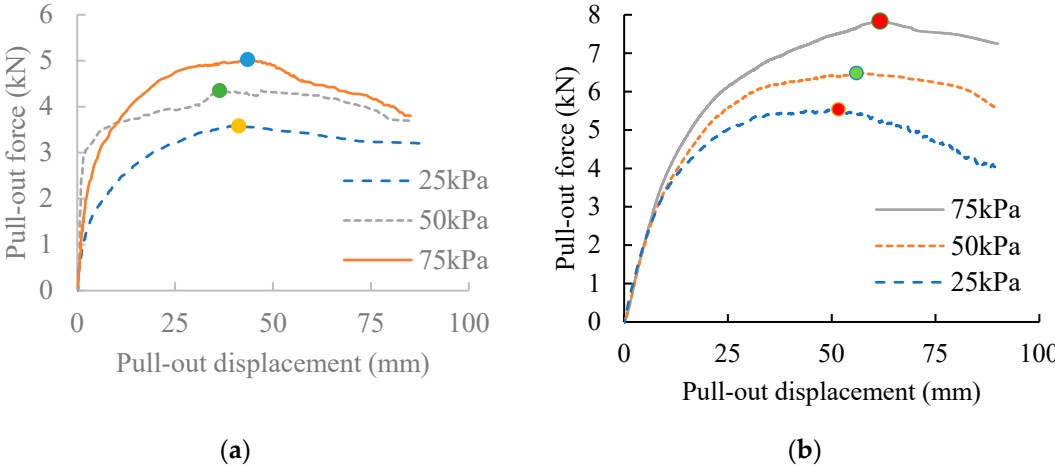

(**a**)　　　　　　　　　　　　　　　　　　　　　　　(**b**)

**Figure 8.** Pull-out force and displacement with silt and gravel. (**a**) Silt; (**b**) gravel.

The fitting curves of normal stress and shear stress for different fillers are shown in Figure 9. In terms of particle size, the silt was the smallest, the sand took the second place and the largest particle size was the gravel. When the geogrid was fully paved in the test box, in which the ribs in both directions were not damaged, the embedded length was 60 cm. Under the condition of 50 kPa normal stress, the maximum pull-out force in silt was 4.350 kN and in sand it was 4.811 kN. The maximum pull-out force was increased by over 10%, or 0.461 kN. The maximum pull-out force in gravel was 6.170 kN, which increased by 28%, or 1.359 kN, compared with that of sand.

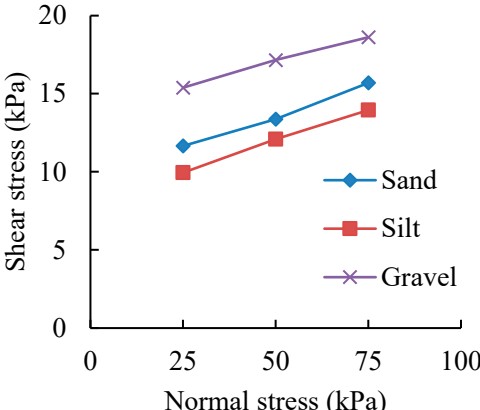

**Figure 9.** Shear and normal stress fitting curve with different materials.

The mechanical performance indicators derived from the fitted curve are shown in Table 7. The results indicated that as the particle size increased, the interface cohesion increased and the interface friction angle decreased. When the filler was sand, the interface cohesion was 9.538 kPa, which was 19% higher than that of silt. The interface cohesion of gravel was 13.822 kPa, which is 44% larger than that of sand. The change in the interface friction angle was not obvious. This is because the larger the particles were, the rougher the particle surface and the more irregular the shapes were. Therefore, the frictional resistance required for mutual displacement and rearrangement of the particles during the pull-out test increased. The roughness of the particle surface causes the increase in friction resistance between the geogrid and the soil. With larger particle size, the mesh's interlocking effect can be fully achieved, and the end resistance between the soil and the geogrid is improved.

When the particle size is large, such as in gravel, the occlusal effect is greater than the friction. Additionally, the more irregular shape of particles is another reason why the shape could enhance the occlusion between particles [23]. In contrast, the friction is larger than the occlusion with silt and sand. Therefore, the shear stress of the geogrid in the gravel was greater than that of the pull-out test in the silt.

**Table 7.** Pull-out test results of silt and gravel.

| Filler | Normal Stress (kPa) | Interface Cohesion (kPa) | Interface-Friction Angle (°) |
|--------|---------------------|--------------------------|------------------------------|
| Silt   | 25 50 75            | 7.982                    | 4.585                        |
| Sand   | 25 50 75            | 9.538                    | 4.614                        |
| Gravel | 25 50 75            | 13.822                   | 3.690                        |

The effects on cohesion result from the combined action of the geogrid and the soil, including the shear resistance of the soil, the frictional resistance between the soil and the geogrid and the pull-out resistance of the geogrid, which significantly improves the strength of the reinforced composite. This would show a cohesion-like improvement in the test result. As for interface-friction angle, the results from the pull-out test were smaller than that of the direct shear test. Moreover, some of the cross-machine ribs of the geogrid were torn during the test, which could have reduced the resistance force of the cross-machine rib, so the interface-friction angles from test results may have been affected.

## 4. The Effect of Particle Diameter on the Friction Characteristics of the Interface between Reinforcement and Soil

In order to gain further detailed insights into the geogrid–soil interaction under pull-out loads, the discrete element software Particle Flow Code (PFC$^{2D}$) has been used in this study. PFC$^{2D}$ utilizes rigid entities (particles and walls) and soft contacts in the numerical modeling. Newton's second law of motion is used to update the positions of the rigid entities due to the forces acting on the soft contacts. The contact forces are then updated based on the force-displacement law. The cycle of the above two successive processes stops when the forces or displacements of the rigid entities reach the expected values in the DEM investigations. Despite the limitations of 2D numerical modeling, the fundamental interface behavior between geogrid and soil can be obtained using PFC$^{2D}$ with reasonable computational time [24].

### 4.1. DEM Model

(1)  Soil

The aim of this study was to reveal how particle size affects the interface characteristics between geogrid and soil, and the particle size of gravel is larger and allows easier observation. As for silt and sand, the particle size is too small, even when its size is doubled. If the particles of silt or sand become large enough, the material might become gravel. So, gravel was chosen for DEM simulation.

To reveal the effect of different particle sizes of gravel on the pull-out test results, the direct shear test model was established by PFC$^{2D}$ based on the laboratory test. Four sets of pull-out test numerical simulations were carried out, and the applied normal loads were 25 kPa, 50 kPa, 100 kPa and 150 kPa. The simulation diagram is shown in Figure 10, and the calibrated gravel meso-parameters are shown in Table 8.

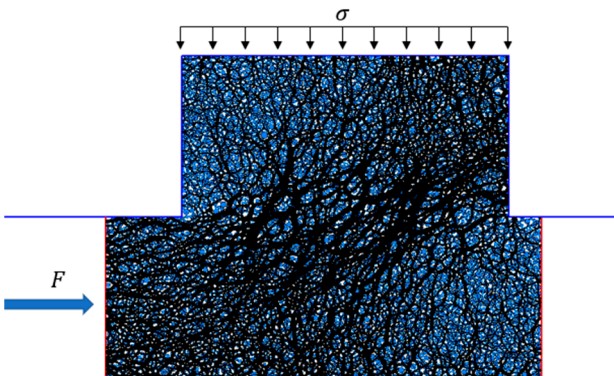

**Figure 10.** DEM diagram.

**Table 8.** Gravel meso-parameters.

| Particle Density (kg·m$^{-3}$) | Porosity | Coefficient of Friction | Normal Contact Stiffness (N·m$^{-1}$) | Tangential Contact Stiffness (N·m$^{-1}$) |
|---|---|---|---|---|
| 2650 | 0.15 | 0.45 | $1 \times 10^9$ | $1 \times 10^9$ |

(2)　Geogrid

Under normal circumstances, the tensile force obtained in the geogrid tensile test has a nonlinear relationship with the strain of the geogrid, but most of the simulations showed it simplified to a linear relationship. Therefore, based on the parallel bonding model, a linear model was established to represent the strength properties of the geogrid with a linear force–strain relationship in the numerical simulation [24–26].

As shown in Figure 11, the geogrid in the simulated tensile test was composed of 28 particles with a total length of 180 mm. Two 15-mm-diameter particles were used to simulate the geogrid node. The rib on the machine direction was simulated by 3-mm-diameter particles. The DEM parameters on the geogrid are shown in Table 9.

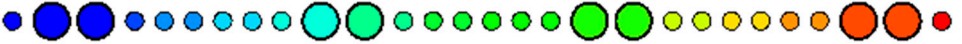

**Figure 11.** DEM simulation on the geogrid.

**Table 9.** DEM parameters on geogrid.

| Items | Index |
|---|---|
| Relative density of particles | 905 |
| Coefficient of friction | 0.3 |
| $pb\_kn$/(GPa·m$^{-1}$) | 6.5 |
| $pb\_ks$/(GPa·m$^{-1}$) | 6.5 |
| $pb\_ten$/(GPa·m$^{-1}$) | 6.5 |
| $pb\_coh$/(GPa·m$^{-1}$) | 6.5 |
| $kn$/(N·m$^{-1}$) | 1e8 |
| $ks$/(N·m$^{-1}$) | 1e8 |

In the numerical simulation of the pull-out test, the filler particles were generated by the layered compaction method, divided into five layers and the method was the same as that for the direct shear test numerical simulation particles. In order to study the effect of filler particle size on the friction characteristics of the interface between reinforcement and soil, the original gravel particles were enlarged and reduced and the initial state of the pull-out test on the original gravel particles was simulated, as shown in Figure 12. The displacement and speed were set to zero, the constant pull-out rate set to 2 mm/min and

the normal stresses were 25 kPa, 50 kPa and 75 kPa. The comparison between the simulated curve and the test result is shown in Figure 13.

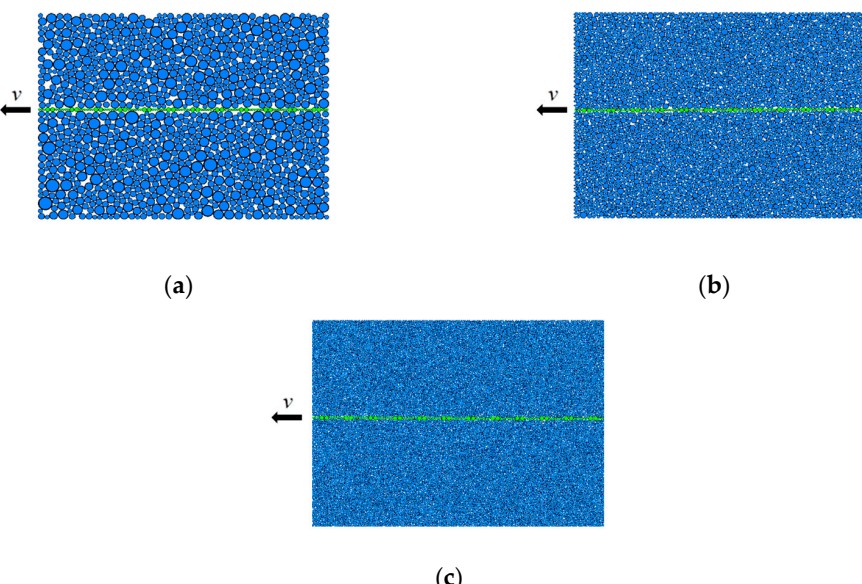

**Figure 12.** The initial state of the model. (**a**) Filler particles double-enlarged; (**b**) original gravel particles; (**c**) particle size reduced by half.

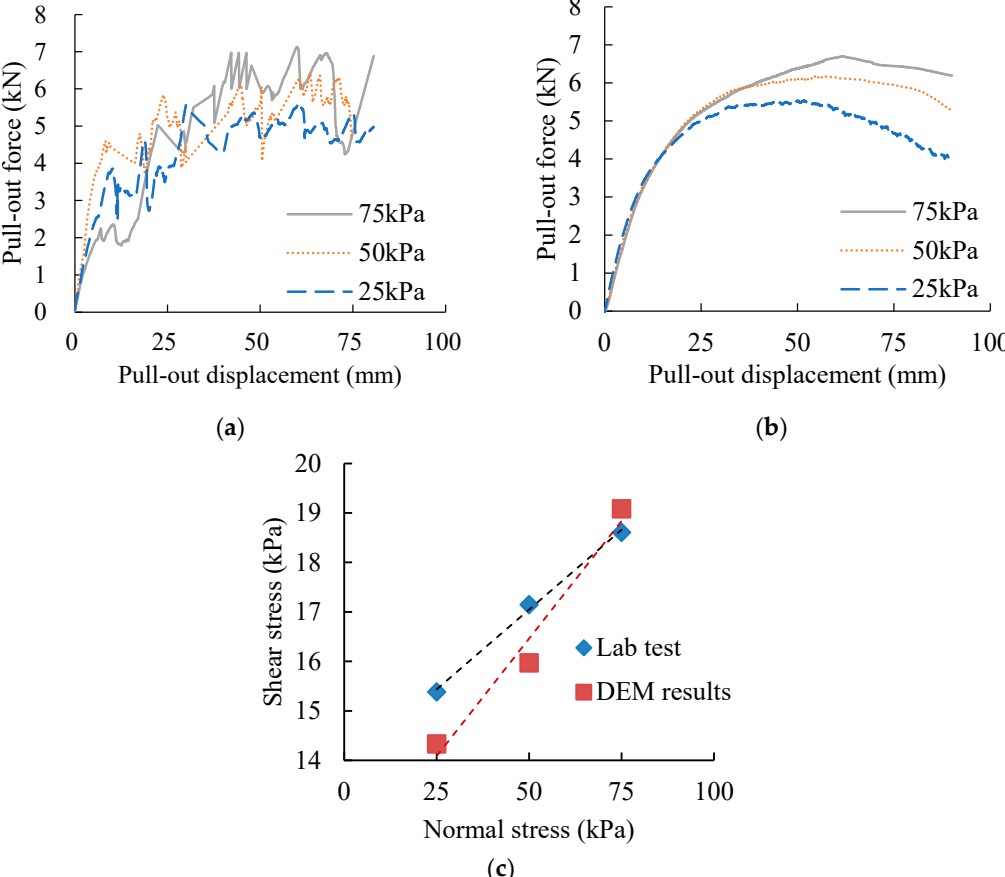

**Figure 13.** Calibration results. (**a**) Curve of pull-out test by DEM; (**b**) curve of pull-out test in laboratory; (**c**) curve of shear and normal stress in pull-out test.

In the DEM model, the dispersion of the filler particles was fully reflected and meso-scaled; so, the curve was not as perfect as the laboratory test, but the results showed the same trend and the shear stress was not much different, so it could be determined that the calibration parameters met the requirements.

*4.2. The Effect of Particle–Mesh Size Ratio on the Friction Characteristics of the Interface between Reinforcement and Soil*

The mesh size of the biaxial geogrid was 32 mm × 33 mm. As 2D software was used, the mesh size of the grid was simplified to be 30 mm × 30 mm, which was defined as the equivalent mesh size to the 30 mm mesh. The reinforced area of the geogrid remained unchanged when the particle size changed, in order to eliminate any influence on the results. On the assumption that the area of reinforcement remains unchanged, changing the particle size of the filler particles changes the contact area between the grid and the filler, resulting in a change in friction. To completely eliminate the effect of changing friction on the test results, the coefficient of friction was set to zero. At this time, the pull-out force was the embedded force between the geogrid ribs and the filler. Under this premise, the influence of filler particle size on the reinforcement characteristics of the geogrid was explored. At the same time, the particle–pore ratio was defined as the ratio of the average particle size of the filler to the equivalent mesh size of the geogrid, as shown in Equation (1).

$$i = \frac{d_{50}}{L_a}. \tag{1}$$

The particle size ranged from small to large; at $d_{50}$ of 2.38 mm, 4.76 mm and 9.52 mm, the ratios of grain to hole were 0.079, 0.159 and 0.317, respectively.

The DEM results showed that the relationship between the pull-out force and the displacement of each test was basically the same. At the beginning of the model test, the drawing force increased rapidly, then it peaked at a certain value with the increase in the displacement, which was the maximum drawing force. Moreover, the pull-out force increased with the normal stress and the particle size.

The pull-out force increases with the increase in normal stress. When the particle–mesh ratio was 0.159, the normal stress increased from 25 kPa to 75 kPa, and the pull-out force increased from 5.1 kN to 6.87 kN; when the particle–mesh ratio was 0.079, the tensile force increased, the pull-out force ranged from 1.87 kN to 4.05 kN; when the particle–mesh ratio was 0.317, the pull-out force ranged from 6.1 kN to 8.83 kN.

Table 10 shows the test results with different particle–mesh ratios. The strain softening appeared after the pull-out force reached the peak. The strain softening inflection point appeared latest when the ratio was 0.317. This indicates that the larger the particle–mesh ratio, the greater the interfacial shear stress.

**Table 10.** Results of simulated pull-out test on the interface between reinforcement and soil.

| Particle–Mesh Size Ratio | Pull-Out Rate (mm·min$^{-1}$) | Normal Stress (kPa) | Maximum Pull-Out Force (kN) | Maximum Pull-Out Displacement (mm) | Shear Strength (kPa) |
|---|---|---|---|---|---|
| | | 25 | 5.16 | 35 | 14.33 |
| 0.159 | 2 | 50 | 5.75 | 45.1 | 15.97 |
| | | 75 | 6.87 | 47.6 | 19.08 |
| | | 25 | 1.87 | 26.3 | 5.19 |
| 0.079 | 2 | 50 | 2.36 | 34.7 | 6.56 |
| | | 75 | 4.05 | 56.3 | 11.25 |
| | | 25 | 6.1 | 47.5 | 16.94 |
| 0.317 | 2 | 50 | 7.77 | 47.7 | 21.58 |
| | | 75 | 8.83 | 52.9 | 24.53 |

The parameters of the reinforcement–soil interface obtained from the simulated pull-out test are listed in Table 11.

**Table 11.** Strength parameters on the reinforcement–soil interface with different particle–mesh ratios.

| Particle–Mesh Ratios | Cohesion (kPa) | Tangent of Internal-Friction Angle | Internal-Friction Angle (°) |
| --- | --- | --- | --- |
| 0.159 | 11.713 | 0.575 | 29.899 |
| 0.317 | 13.435 | 0.821 | 38.369 |
| 0.079 | 1.611 | 0.428 | 23.171 |

As the particle–mesh ratio increased, the cohesion of the reinforced sample continued to increase. The cohesion gradually increased from 1.611 kPa to 13.435 kPa as the particle–mesh ratio changed from 0.079 to 0.317. The cohesion rose with increasing particle size. The internal-friction angle of each sample also increased with the increase in particle size. This indicates that the contact area between the geogrid and the filler will increase with larger particle size, resulting in the change of shear stress.

In different particle–mesh ratios, the relationship between normal stress and interface-friction coefficient is shown in Figure 14. It can be seen from Figure 15 that as the normal stress decreased, the friction coefficient gradually increased. When the particle–mesh ratio was 0.159, it decreased from 0.573 at 25 kPa to 0.254 at 75 kPa. This conclusion was the same as that from the study of the interfacial-friction coefficient under the laboratory pull-out test. Under the condition of 25 kPa normal stress, the interface friction coefficients were 0.208, 0.573 and 0.678 when the ratios were 0.079, 0.159 and 0.317, respectively. They changed to 0.131, 0.319 and 0.432 when the normal stress was 50 kPa, and 0.150, 0.254 and 0.327 when the normal stress was 75 kPa.

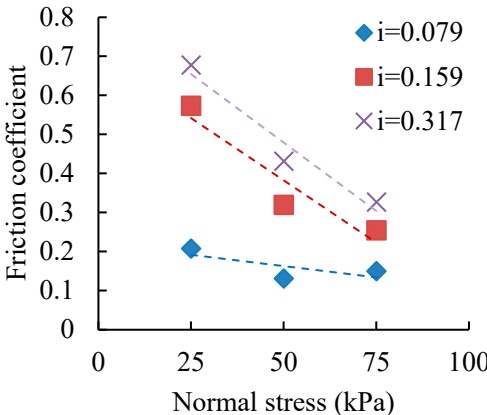

**Figure 14.** Interface-friction coefficient under different normal stresses.

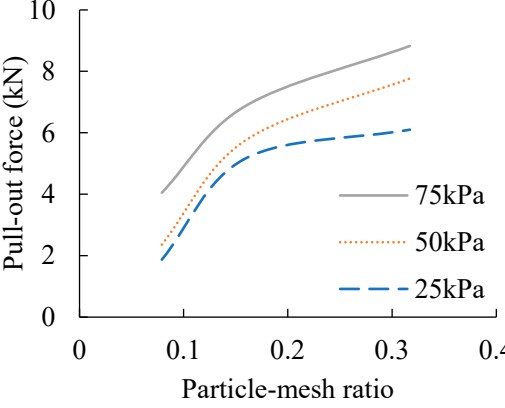

**Figure 15.** The effect of particle–mesh ratio on the maximum pull-out force.

The maximum pull-out force and the displacement under different particle–mesh ratios are shown in Figures 15 and 16.

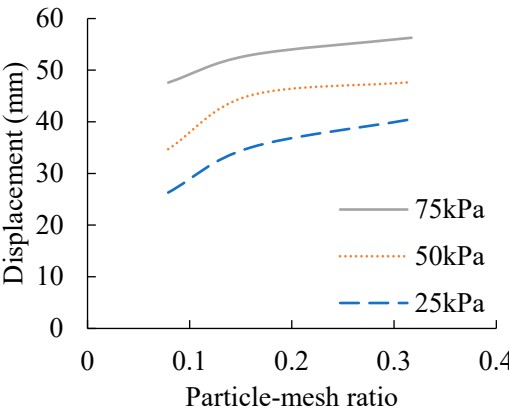

**Figure 16.** The effect of particle–mesh ratio on displacement.

The particle–mesh ratio had a certain influence on the maximum pull-out force and displacement. The maximum pull-out force increased with larger particle–mesh ratio, but the corresponding displacement flattened after the rise. This indicates that the reinforcement effect was best when the ratio was 0.159. It also shows that there was a matching relationship between the mesh size of the geogrid and the particle size of the soil. In order to achieve a better performance of the reinforced structure, both particle size and geogrid mesh should be selected.

Overall, these results will help designers with choosing the geogrid type based on the materials used in construction, such as larger mesh that could be used in mountainous areas where gravel is normally used and smaller mesh geogrids for reinforced structures in coastal areas. This could provide a reference to improve the rational design of reinforced structures. Still, more laboratory tests and DEM analyses on particle–mesh ratio are needed to provide a more accurate reference on reinforced structure design, such as geogrid-reinforced walls.

## 5. Conclusions

The influence of normal stress on the friction characteristics of the interface between the reinforcement and soil was analyzed, together with the characteristics of the interface between the reinforcement and soil in different fillers (gravel, sand and silt). The main conclusions are as follows:

(1) Under normal static load, laboratory pull-out tests obtained the strength indexes of reinforced soil with different particle sizes. The strength characteristics and reinforcement effect indicated that, after reinforcement, the internal-friction angle remained the same and cohesion increased significantly, and the samples showed strain softening during the test procedure.

(2) The change in normal stress had a significant effect on the results of the pull-out test. The pull-out force increased by 14.6% at least with greater normal stress. It indicates that overlaying load is a key factor for the stabilization of reinforced structure. Additionally, the friction coefficient with 25 kPa normal stress is 44.8% of that with 75 kPa.

(3) The larger the particle size of the filler, the more likely it was to rotate and move during the pull-out test, resulting in the mutual displacement of soil particles at the interface between the reinforcement and the soil, thereby facilitating an increase in the tensile resistance of the specimen. The greater the interlocking force, the greater the pull-out force. However, this conclusion was limited by the test materials, and it cannot explain the friction characteristics between the geogrid and the soil when the particle size of the filler is larger than the geogrid mesh size.

**Author Contributions:** Conceptualization, G.Y. and Y.Z.; methodology, Y.Z.; software, Z.W.; validation, Z.W. and Y.Z.; formal analysis, Y.Z.; investigation, Y.Z.; resources, S.Y. and Z.W.; data curation, Y.Z.; writing—original draft preparation, Y.Z.; writing—review and editing, G.Y. and S.Y. All authors have read and agreed to the published version of the manuscript.

**Funding:** This research was funded by the National Natural Science Foundation of China, grant number 51378322 and 51709175, and the National Science Foundation of Hebei Province, grant number E2017210148, E2018210097 and E2019208159.

**Conflicts of Interest:** The authors declare no conflict of interest.

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
