# Peer review of "Research on the Effect of Particle Size on the Interface Friction between Geogrid Reinforcement and Soil"

_sustainability, doi:10.3390/su142215443_

Round 1

Reviewer 1 Report (New Reviewer)

In this manuscript, pull-out tests were conducted with different fillers, such as sand, silt and gravel by using a self-developed large-scale pull-out testing machine. The pattern of how the soil particles moved during the pull-out test was revealed with discrete element method. The idea of this manuscript is reasonable, but there are still some defects. This manuscript can be accepted after major revision:

(1)    In the introduction, the author lists a large number of previous research results. However, this only indicates that the authors may have reviewed these literature earlier. Readers would like to see the author's evaluation of these literature and the innovative points of this manuscript. The author should revise the introduction.

(2)    In section 2.3.1, where were these test materials sampled from? Figure 1 is of low quality and is recommended to be redrawn.

(3)    How was the maximum dry density and minimum dry density measured during compaction test? It is recommended that the authors read the descriptions of the physical parameters of the test materials in the following literature and modify them accordingly:

Particle breakage mechanism and particle shape evolution of calcareous sand under impact loading. Bulletin of Engineering Geology and the Environment. 2022, 81. DOI: 10.1007/s10064-022-02868-9.

(4)    In line 120-124, I admit that's a reasonable explanation. However, these explanations are not the experimental phenomena observed in the experiment, but the speculation of the authors based on the experimental phenomena. It is suggested that the authors cite similar research literature for these explanations, and then demonstrate their reliable.

(5)    The peaks in Figs. 4 and 8 are mentioned in the analysis section. It is suggested that the author should mark these peaks in the Figs. 4 and 8.

(6)    In line 188-189, the larger the particle size, the stronger the occlusal effect between particles? Is it because the shape of coarse particles is more irregular?The following literature can support this statement:

Shape characteristics of coral sand from South China Sea. Journal of Marine Science and Engineering. 2020, 8(10): 803. DOI:10.3390/jmse8100803.

(7)    In line 222-224, in which research results, the geogrid tensile test has a linear relationship with the strain of the geogrid? The authors should explicitly point out these similar studies, otherwise the simulation conclusions cannot be favourably supported.

(8)    In line 288-289,’ The cohesion rose with increasing particle size. The internal-friction angle of each sample also increased with the increase in particle size.’ The author should briefly explain what causes this result.

(9)    In line 300-301,’This indicates that the smaller particle sizes had a larger impact on geogrid–soil interface characteristics.’ Discrete element method can replace test to some extent. However, the result obtained by the discrete element method is the same as the test result, which is only a phenomenon. The results obtained by the discrete element method cannot be directly regarded as the mechanism leading to some phenomenon. The results obtained by the author based on the discrete element method still need to explain its internal mechanism.

(10) In line 326-329, it is suggested that the author rewrite this part of conclusions with quantitative indicators.

Author Response

Thanks very much for taking your time to review this manuscript. I really appreciate all your comments and suggestions! Please find my itemized responses in below and my revisions/corrections in the re-submitted files. Please see the attachment.

Reviewer 2 Report (Previous Reviewer 1)

It is considered to be a paper analyzing the friction behavior in soil structures to which geosynthetics were applied.

1. In fact, if the discrete element method (DEM) analysis result applied in this paper is applied to the construction site, please add a description of the reliability of how much it corresponds to the frictional behavior of the soil and geogrids at the actual construction site.

2. If there is a reference or citation performance that was analyzed by applying the discrete element method (DEM) analysis, please introduce the validity of DEM in connection with the friction behavior analysis of this paper.

3. Since the particle distribution of filling materials is not a filling material used at the construction site, damage to the geogrid may occur due to the angularity of the soil used at the actual construction site. In this case, the reduction factor due to damage during construction affects the design strength and long-term performance of the geogrid. This paper does not contain any related content, but please add an explanation regarding the change in the frictional properties of the geogrid.

4. Please quantitatively supplement the "conclusion" part based on the experimental results of this paper.

5. Please rewrite the references in the order of the test results, and write mainly references directly cited.

Author Response

Thanks very much for taking your time to review this manuscript. I really appreciate all your comments and suggestions! Please find my itemized responses in below and my revisions/corrections in the re-submitted files. "Please see the attachment.

Reviewer 3 Report (New Reviewer)

Review Comments

The paper represents “Research on the effect of particle size on the interface friction
between geogrid reinforcement and soil
”.  Authors have investigated the effects of the difference in interfacial friction properties between various particle-size materials through the laboratory tests and DEM analysis. However, the previous literature related to this area, research gap, major goals and contribution of the conducted work was not properly discussed in the manuscript. The authors should submit a very thoroughly revised version, addressing the comments mentioned in below:

1.     The authors should explain elaborately the research gap, major goals, novelty of the work, objectives of the study and contribution of the conducted work in the introduction section. Also include more literature (especially recent) related to this field.

2.     Check the unit of Sample volume (cm3) in Table 1. This should be like cm3.

3.     The optimum moisture content of sand is not mentioned in Table 2. What is the reason behind that?

4.     The procedure of the pull-out test of the reinforcement–soil interface is not clearly mentioned. What is the basis of selection of the test box dimension? Is there any scaling law used for this study? If yes, then mention in details. Also mention the procedure of soil bed preparation in the test box for each type of soil. The following papers can be referred for soil bed preparation:

DOI: https://doi.org/10.1061/(ASCE)GT.1943-5606.0002176

DOI: https://doi.org/10.1007/s10706-015-9875-7

DOI: https://doi.org/10.1016/j.oceaneng.2022.112139

DOI: https://doi.org/10.1080/1064119X.2020.1766607

5.     It is suggested to improve the conclusion and include the value based results in conclusion part. 

6.     It is suggested to include the limitation and future prospective of this study.

7.     English language should be improved throughout the manuscript.

Author Response

Thanks very much for taking your time to review this manuscript. I really appreciate all your comments and suggestions! Please find my itemized responses in below and my revisions/corrections in the re-submitted files. "Please see the attachment.

Round 2

Reviewer 1 Report (New Reviewer)

The author has replied to all my questions and made a lot of revisions to the previous manuscript. Now, I am willing to accept this manuscript in present form.

Reviewer 3 Report (New Reviewer)

Accept in current form.

This manuscript is a resubmission of an earlier submission. The following is a list of the peer review reports and author responses from that submission.

Round 1

Reviewer 1 Report

This paper is considered to be a popular paper that analyzes the friction characteristics between the filler material and the geogrid, which is a reinforcement material.

Please correct the following information and provide additional explanation.

1. For “Means of discrete element numerical simulation”,

    i) To be needed a specific explanation of the definition

    ii) What is the purpose of applying this theory to the experimental data             analysis?

  iii) Please add a quantitative analysis of how the application of this theory          affects the frictional properties of the nodular filler and the geogrid.

2. Please clearly state the connection between the reference and the content     cited in the corresponding text.

3. Please edit and write the content of the conclusion by including it in the       abstract.

Author Response

Response to Reviewer 1 Comments

Thanks very much for taking your time to review this manuscript. I really appreciate all your comments and suggestions! Please find my itemized responses in below and my revisions/corrections in the re-submitted files.

Thanks again!

  1. For “Means of discrete element numerical simulation”,
  2. i) To be needed a specific explanation of the definition
  3. ii) What is the purpose of applying this theory to the experimental data analysis?

iii) Please add a quantitative analysis of how the application of this theory affects the frictional properties of the nodular filler and the geogrid.

Response 1: Thanks for the comment! The relating sentences in the abstract has been changed to “Finally, to reveal the pattern of how the soil particle moved during pull-out test from the microscopic view and the effect on particle-mesh size ratio, a series of discrete element method (DEM) analysis were conducted by PFC2D.”

  1. Please clearly state the connection between the reference and the content cited in the corresponding text.

Response 2: Thanks for the comment! The purpose of this study has been add on the last paragraph of Introduction, which is “Currently, researches on friction characteristics of reinforced soil were based on the laboratory tests, and the test results such as cohesion and friction angle were macroscopic. Also, most of them focused on fine materials like sand, the difference of interfacial friction properties between various particle-size materials was rare to seen. So based on the laboratory test and DEM analysis, the internal friction characteristics of reinforced soil under different particle-size materials was studied. Moreover, it may provide references to the future design and application on GRS structures.”

  1. Please edit and write the content of the conclusion by including it in the abstract.

Response 3: Thanks for the comment! The conclusions have been summarized in the abstract, which has been changed to “According to the test results, the greater the stress applied in the normal direction, the greater the maximum pull-out force. As for the different fillers, shear stress from larger particle size like gravel was larger than that of sand and silt. Finally,  to reveal the pattern of how the soil particle moved during pull-out test from the microscopic view and the effect on particle-mesh size ratio, a series of discrete element method (DEM) analysis were conducted by PFC2D. The results indicated that larger particle is more likely to rotate and move during the test, and makes the interlocking effect greater between the geogrid and soil, which lead to larger pull-out force in the laboratory test.”

Reviewer 2 Report

Part1.

1.   In the Figure 1, I think Figure1(a) is curve of the particle size distribution of gravel. And Figure1(c) is curve of the particle size distribution of silt. So, you have change the figure1 titles and further check all titles of figures and tables. Also, you can change figure1(c) title that sandy silt.

2.   In the table4, used the wrong terms that medicine direction and cross medicine direction. 

(Terms: ‘machine direction (MD)’, ‘cross machine direction (CMD)’) 

3.   In the Figure 4, 5 and figure 8 (a, b), you must write the title with name of the filler material used.

4.   In the table8, write unit of particle density.

Part2. 

1.   In the table2, what is the optimal moisture content value of sand in compaction test? 

2.   In the table3, friction angle of sand is 26.94, so it was loose sand? Also, filler material silt that containing clay fraction(0.075mm-0.002mm) and sand fraction. So, when additional clay fraction in sand mixture, increases the internal angle of friction, and decrease cohesion. but direct shear test results of this study, friction angle of silt was so lower, why? 

3.   In the figure 6, friction angle value from pull-out test didn’t match the figure6.

4.   Combine the tables 5and 6 for comparison.

5.   In part 4, why did you choose only gravel for DEM model simulation?

Part3. 

1.   In the row 175, resulting in the pull-out force increased with the increase of normal stress. (Of course, this is right immobilization effect.) But in row 176, ‘It can also be seen that the pull-out force increases with the increase of displacement’ (I think it is not immobilization effect and maybe wrong judgment. Because in the test result, ‘the break was set when the pill-out displacement reached 90mm, and look at figure4, the pull-out force increased when they reach 48.149mm, 53.946mm and 62.101mm of pull-out displacement, then they decreased again. So, maybe figure5 is meaningless)

2.   In the table 5 and 6, under the same experimental condition (as use same geogrid, under the same normal stresses) cases, larger particle size material has a higher friction angle than small particle size material. But in this experiment, gravels friction angle was lower, reason?

3.   And particle size of sand and silt were different, but why did get almost same value of friction angle after being reinforced? Is that reason being only roughness of particle surface?

4.   Why is there cohesion from tests using sand and gravel? Also, the internal friction angles are too small value in all pull-out tests, its possible?

Author Response

Response to Reviewer 2 Comments

Thanks very much for taking your time to review this manuscript. I really appreciate all your comments and suggestions! Please find my itemized responses in below and my revisions/corrections in the re-submitted files.

Thanks again!

Part1.

  1. In the Figure 1, I think Figure1(a) is curve of the particle size distribution of gravel. And Figure1(c) is curve of the particle size distribution of silt. So, you have change the figure1 titles and further check all titles of figures and tables. Also, you can change figure1(c) title that sandy silt.

Response 1: Thank you for the comment! The figure title has been changed.

  1. In the table4, used the wrong terms that medicine direction and cross medicine direction.

Response 2: Thanks for the comment! All the wrong words have been carefully checked and changed.

  1. In the Figure 4, 5 and figure 8 (a, b), you must write the title with name of the filler material used.

Response 3: Thanks for the comment! All the figure title has been corrected.

  1. In the table8, write unit of particle density.

Re: Thanks for the comment! The particle density in Table 8 is “Relative density of particles”, which is dimensionless.

Part2.

  1. In the table2, what is the optimal moisture content value of sand in compaction test?

Response 1: Thanks for the comment! As for the sand, there is no optimal moisture content value, the maximum and minimum dry density were 1.83 g•cm-3 and 1.51 g•cm-3.

  1. In the table3, friction angle of sand is 26.94, so it was loose sand? Also, filler material silt that containing clay fraction(0.075mm-0.002mm) and sand fraction. So, when additional clay fraction in sand mixture, increases the internal angle of friction, and decrease cohesion. but direct shear test results of this study, friction angle of silt was so lower, why?

Response 2: Thanks for the comment! The friction angle of sand was compacted and tested by diret-shear test. The friction angle of reinforced materials should remained almost same than the origin materials. As the friction angle in the manuscript is the friction angle between geogrid and soil, not the characteristic of soil.

  1. In the figure 6, friction angle value from pull-out test didn’t match the figure6.

Response 3: Thanks for the comment! The fitting curve has been put in figure 6, and all the numbers have been checked.

  1. Combine the tables 5and 6 for comparison.

Response 4: Thanks for the comment! The results from pull-out test with sand have been put in Table 6 for comparison.

  1. In part 4, why did you choose only gravel for DEM model simulation?

Response 5: Thanks for the comment! Because the aim of manuscript is to study how the particle size affect the interface characteristics between geogrid and soil, and the particle size of gravel is larger and easier for observation. As for silt and sand, the particle size is so small; even with twice enlarge of its size. And if particle of silt or sand became large enough, the material might became gravel. So gravel has been chosen for DEM simulation.

Part3.

  1. In the row 175, resulting in the pull-out force increased with the increase of normal stress. (Of course, this is right immobilization effect.) But in row 176, ‘It can also be seen that the pull-out force increases with the increase of displacement’ (I think it is not immobilization effect and maybe wrong judgment. Because in the test result, ‘the break was set when the pill-out displacement reached 90mm, and look at figure4, the pull-out force increased when they reach 48.149mm, 53.946mm and 62.101mm of pull-out displacement, then they decreased again. So, maybe figure5 is meaningless)

Response 1: Thanks for the comment! This is a normal test phenomenon. The occlusal effect between the geogrid and the soil particles lead to the increase of the pulling force in the early stage of the test. When the displacement keep increasing, the soil particles start to rotate and move, then the occlusal effect between the geogrid and the soil particles decreases, and most of the pull-out force is friction between geogrid and soil, so there is a decrease in the pull-out force after reaching the peak.

  1. In the table 5 and 6, under the same experimental condition (as use same geogrid, under the same normal stresses) cases, larger particle size material has a higher friction angle than small particle size material. But in this experiment, gravels friction angle was lower, reason?

Response 2: Thanks for the comment! This phenomenon is related to the particle size of soil particles and the grid size of the geogrid. When the particle size is large, like gravel, the occlusal effect is greater than friction. Otherwise, the friction is larger than occlusion with silt and sand.

  1. And particle size of sand and silt were different, but why did get almost same value of friction angle after being reinforced? Is that reason being only roughness of particle surface?

Response 3: Thanks for the comment! The roughness is one reason that the friction angle is almost the same, the other reason is that the matching degree of particle size of soil particles and the grid size of the geogrid. If the particle size is large, like gravel, the particle could be embedded between grids of geogrid, and the occlusal effect would be greater than friction.

  1. Why is there cohesion from tests using sand and gravel? Also, the internal friction angles are too small value in all pull-out tests, its possible?

Response 4: Thanks for the comment! The aim to test cohesion of sand and gravel is to compare the unreinforced and reinforced sample, and reveal the interface cohesion of geogrid-reinforced sand/gravel.

Reviewer 3 Report

Authors investigate the effect of particle size on the interface friction between geogrid reinforcement and soil. The study cannot offer any new method whilst some useful data is generated for the use of designers. The experiment and modelling process is described relatively in detail, but the discussion is superficial and vague. It is a coarse work report, not a research article.

1. The components of the test equipment should be marked in the Figure 3.

2. The mechanism figure of the pull-out process should be plotted to vividly describe the interaction between soil particles and geogrid.

3. The logic of the introduction needs to be combed and the core meaning of cited references should be refined concisely and comprehensively. The inspiration, deficiency and connection of the references need to be concluded, which is the reason why they are cited. The introduction is not to list others’ work simply. It’s the meaning of this work. So where is it? It should be elaborated in the introduction!

4. The writing must be improved and the grammar and format have to be checked carefully, which has influenced the understanding.

Author Response

Response to Reviewer 3 Comments

Thanks very much for taking your time to review this manuscript. I really appreciate all your comments and suggestions! Please find my itemized responses in below and my revisions/corrections in the re-submitted files.

Thanks again!

  1. The components of the test equipment should be marked in the Figure 3.

Response 1: Thanks for the comment! All the components of the test equipment have been shown and marked in Figure 3(b).

  1. The mechanism figure of the pull-out process should be plotted to vividly describe the interaction between soil particles and geogrid.

Response 2: Thanks for the comment! The mechanism figure of the pull-out process has been shown in Figure 3(a).

  1. The logic of the introduction needs to be combed and the core meaning of cited references should be refined concisely and comprehensively. The inspiration, deficiency and connection of the references need to be concluded, which is the reason why they are cited. The introduction is not to list others’ work simply. It’s the meaning of this work. So where is it? It should be elaborated in the introduction!

Response 3: Thanks for the comment! The introduction has been rewritten.

The introduction has been changed to “Geosynthetics-reinforced soil walls (GRSW) have been widely used in various engineering fields, such as roads, railways, etc. GRSW is a viable replacement for conventional concrete-retaining structures in infrastructure development and remedial treatments around the world. It is of great importance to investigate the interaction mechanism of geogrid-soil in the design and application of geogrid reinforced soil structures. There are multiple research methods to investigate the interaction of geogrid-soil, and the pull-out test is the most effective one among them. A lot of researches have been conducted on the interaction of geogrid-soil by laboratory test.

Xiao [1] Wang [2] Chen [3] et al. have the influence of different normal pressure, geogrid type and embedded length on the pull-out characteristics analyzed and discussed through the pull-out test of geogrid in sand. Moraci[4], Bisht[5], Alagiyawanna[6] and Teixeira[7] also studied the pull-out characteristics of geogrid in fine-grained soil under different conditions. Wang [8] investigated the monotonic and cyclic shear behavior of the grit-geogrid interface through a series of experiments.

Ding [9] and Tang [10] et al. studied the influence of the mesh size of the geogrid on the interface characteristics through the pull-out test of the geogrid in fine sand. Zhou[11] and Ezzein[12-14] used transparent fillers in a visual model box to study the microscopic mechanism of the interaction between the geogrid and sand.

Zuo [15] and Abdi [16] performed pull-out and direct shear tests of geogrids in sand-gravel and cohesive soils. The results showed that the shear strength of the contact surface between the geogrid and the clay is very low, but the shear strength of the contact surface with the sand-gravel material was higher. Kim [17] conducted large-scale direct shear tests on three types of coarse-grained soils, showing that the larger the particle size, the higher the shear strength.

Currently, researches on friction characteristics of reinforced soil were based on the laboratory tests, and the test results such as cohesion and friction angle were macroscopic. Also, most of them focused on fine materials like sand, the difference of interfacial friction properties between various particle-size materials was rare to seen. So based on the laboratory test and DEM analysis, the internal friction characteristics of reinforced soil under different particle-size materials was studied. Moreover, it may provide references to the future design and application on GRS structures.”

  1. The writing must be improved and the grammar and format have to be checked carefully, which has influenced the understanding.

Response 4: Thanks for the comment! The writing and format have been carefully checked though out the manuscript.

Round 2

Reviewer 3 Report

The manuscript can be accepted.

Author Response

Dear editor,

Thank you for all the work!

  1. The evidences such as past studies having the same finding to support in response to the review comment have been add in references 18 to 21. They have been highlighted to green.

The references 18-21 are as follows:

  1. Zhang, M., Ma, Y., Qiu, C. Influence of Strengthened Node Arrangement on Pull-Out Characteristics of Biaxial Geogrid. Journal of Shanghai Jiaotong University, 2020, 54(12), 1307-1315.
  2. Cao, W., Zheng, J., Zhou, Y. Comparative Experimental Investigation of Geogrid-soil Interface Behavior of Biaxial and Triaxial Geogrid. Journal of Hunan University(Natural Sciences), 2019, 46(1), 109-116.
  3. Shi, D., Liu W., Shui, W., Liang Y. Comparative experimental studies of interface characteristics between uniaxial/biaxial plastic geogrids and different soils. Rock and Soil Mechanics, 2009, 30(8), 2237-2244.
  4. Meng, Z. Study of Shear Properties of Geogrid Reinforced Tailings Sand. Highway Engineering, 2014, 39(1), 250-255.

  1. All the changes based on comments from reviewer 2 have been highlighted in the revised manuscript, as yellow for Part 1, green for Part 2, and blue for Part 3. Also, those changes have been marked with annotation with each comment correspond to.

  1. Moreover, the authorship change form has been filled and the reason to replace the 4th author is that she has quit the research group right after this project began.

Again, thanks for all the work!

Best regards,

Yunfei Zhao
